# In Vivo Anticoagulant and Antithrombic Activity of Depolymerized Glycosaminoglycan from *Apostichopus japonicus* and Dynamic Effect–Exposure Relationship in Rat Plasma

**DOI:** 10.3390/md20100631

**Published:** 2022-10-02

**Authors:** Han Wang, Dandan He, Linlin Duan, Lv Lv, Qun Gao, Yuanhong Wang, Shuang Yang, Zhihua Lv

**Affiliations:** 1Key Laboratory of Marine Drugs, Ministry of Education of China, School of Medicine and Pharmacy, Ocean University of China, Qingdao 266003, China; 2Laboratory of Marine Drugs and Bioproducts, Pilot National Laboratory for Marine Science and Technology, Qingdao 266237, China

**Keywords:** *Apostichopus japonicus*, glycosaminoglycan, antithrombotic, oral anticoagulant, effect–exposure relationship

## Abstract

Glycosaminoglycan from *Apostichopus japonicus* (AHG) and its depolymerized fragments (DAHGs) are anticoagulant fucosylated chondroitin sulfate. The aim of this study was to further evaluate the anticoagulant and antithrombic activity of AHG and DAHGs, as well as reveal the dynamic relationship between exposure and effect in vivo. The results demonstrated that AHG100 (Mw~100 kDa), DAHG50 (Mw~50 kDa), and DAHG10 (Mw~10 kDa) exhibited potent anticoagulant activity by inhibiting intrinsic factor Xase complex (FXase) as well as antithrombin-dependent factor IIa (FIIa) and factor Xa (FXa). These glycosaminoglycans markedly prevented thrombosis formation and thrombin-induced platelet aggregation in a dose- and molecular weight-dependent manner in vitro and in vivo. The further bleeding time measurement indicated that DAHG10 exhibited obviously lower hemorrhage risks than native AHG100. Following oral administration, DAHG10 could be absorbed into blood, further dose-dependently prolonging activated partial thromboplastin time (APTT) and thrombin time (TT) as well as inhibiting FXa and FIIa partially through FXase. Anticoagulant activity was positively associated with plasma concentration following oral administration of DAHG10. Our study proposed a new point of view to understand the correlation between effects and exposure of fucosylated chondroitin sulfate as an effective and safe oral antithrombotic agent.

## 1. Introduction

Thrombotic disease, especially ischemic heart disease and stroke, is one of the major causes of death worldwide, with increasing incidence and mortality [1,2]. The treatment of thrombotic disorders generally requires anticoagulant therapy, which typically starts with parenteral unfractionated heparin (UFH), or low molecular weight heparin (LMWH), followed by subsequent oral warfarin, a vitamin K antagonist, or one of the direct oral anticoagulants, including factor Xa (FXa) inhibitors (rivaroxaban, apixaban, edoxaban) and a direct thrombin (factor IIa, FIIa) inhibitor (dabigatran) [3]. Generally, when long-term anticoagulation is needed, direct oral anticoagulants are preferred over parenteral agents because they do not require regular laboratory monitoring. However, anticoagulants of all types have not yet addressed the risk of bleeding, without exception [4,5,6]. In recent years, many inhibitors, especially carbohydrates, targeting the components of intrinsic coagulation pathways, such as intrinsic factor Xase complex (FXase, factor IXa-VIIIa-Ca^2+^-phospholipid complex), have become a research focus for their low risk and wide therapeutic window [7,8,9].

Holothurian glycosaminoglycan (HGAG), which is also known as fucosylated chondroitin sulfate (FCS), is a sulfated polysaccharide isolated from sea cucumber, and its chemical structure with a backbone of chondroitin sulfate and sulfated fucose branches is unique. Numerous studies have indicated that HGAGs from the sea cucumber *Thelenota ananas*, Ludwigothurea grisea, *Apostichopus japonicus*, and *Holothuria nobilis* possess both potent anticoagulant and antithrombotic activity, and the mechanisms include inhibition of FXa generation by the intrinsic FXase complex and antithrombin III (AT-III)- or heparin cofactor II (HCII)-dependent inhibition of thrombin (FIIa) [10,11,12]. It was reported that oral administration of HGAG and its depolymerized component (DHG) exhibited noteworthy anticoagulant capacity, and its side effect of hypotension is avoided via oral administration [13,14]. Preliminary pharmacokinetics revealed that 0.1% of DHG with an average molecular weight (Mw) of 12 kDa was traced through urine 24 h after oral dosing at 50 mg/kg [15]. Therefore, it is hypothesized that orally administered HGAG could be absorbed through the gastrointestinal tract into circulation to exert its anticoagulant effect. However, no direct link to clarify the relationship between quantitative structure and dynamic processes in activity has been reported, and the information regarding this relationship is important to establish bio-accessibility as anticoagulants for treating thrombotic diseases.

In previous studies, the anticoagulant properties of a fucosylated chondroitin sulfate, which is referred to as glycosaminoglycan, from *Apostichopus japonicus* (AHG, Figure 1), and its depolymerized fragments (DAHGs) were demonstrated [16]. Further study revealed that AHG and DAHGs could be absorbed and transported across the Caco-2 cell monolayer and M cell model via endocytosis [17]. It is worth noting that the lower molecular weight forms were more readily absorbed.

The aim of this study was to evaluate the anticoagulant and antithrombic activity of AHG and DAHGs and reveal the dynamic relationship between exposure and effect in vivo. Here, native polysaccharide with the weight average Mw of ~100 kDa (AHG100) and its depolymerized derivatives DAHG 50 (Mw~50 kDa) and DAHG10 (Mw~10 kDa) were investigated. Their effects on coagulation (co)factors and platelet aggregation in vitro were determined. Then, the anticoagulant and antithrombotic activity together with the corresponding bleeding risks were comprehensively evaluated. Additionally, their efficacy on anticoagulation was revealed in rats after oral administration. The correlation between effect and exposure was further studied by dynamically describing the inhibition of FIIa and FXa activity and plasma concentration. Our study provides a framework and evidence in chief for development of effective oral anticoagulants for the treatment of thrombotic diseases based on sulfated polysaccharides from marine organisms.

## 2. Results

### 2.1. Characterization of AHG100 and DAHGs

As shown in Table 1 and Appendix A, AHG100 (Mw 94.16 kDa), DAHG50 (Mw 51.32 kDa), and DAHG10 (Mw 9.97 kDa) were obtained with the molar ratios of β-D-glucuronic acid (GlcA), β-D-acetylgalactosamine (GalN), and α-L-fucose (Fuc) mirroring each other among all peaks. The structural consistency of intact AHG and DAHGs was further confirmed by the ^1^H nuclear magnetic resonance (NMR) (Figure 2A) and Fourier transform infrared (FT-IR) spectra (Figure 2B). The detailed peaks and absorption bands have been assigned according to our previous reports [16]. All the data indicated that the primary structure of these products was similar to those of native AHGs. The results of the study add to the growing body of evidence that no major functional group transformations were found during the peroxide depolymerization [16,18,19].

### 2.2. Inhibitory Effects of AHG100 and DAHGs on Coagulation (Co)Factors In Vitro

As shown in Figure 3A–C, the inhibitory effect of depolymerized products on intrinsic FXase complex, FXa, and FIIa depends on the Mw and the concentration. In particular, AHG100 and DAHG50 demonstrated considerably more effective inhibitory effects on the intrinsic FXase complex with the concentration inhibiting 50% (IC_50_) of 0.13 and 0.46 μg/mL, respectively, compared with UFH (0.78 μg/mL). When Mw was reduced to 9.97 kDa, considerable suppression was still observed (IC_50_ 2.19 μg/mL), with effects even stronger than that of LMWH (IC_50_ 28.12 μg/mL). Moreover, both native AHG100 and its depolymerized products DAHG50 and DAHG10 enhanced FXa inactivation by AT-III, and the IC_50_ was 17.54, 24.77, and 71.19 μg/mL, relatively. The inhibitory effect decreased slightly with a reduction in molecular size. The AT-III-mediated anti-FIIa inhibition effect was also concentration-dependent. However, AT-III-dependent anti-FIIa activities visibly weakened as Mw decreased. DAHG10 (IC_50_ 26.89 μg/mL) was much weaker than native AHG100 (IC_50_ 1.11 μg/mL) and DAHG50 (IC_50_ 1.14 μg/mL).

### 2.3. Effect of AHG100 and DAHGs on Thrombin-Induced Platelet Aggregation

Platelets are a key element in the process of physiologic hemostasis and artery thrombosis formation [20]. Therefore, we examined thrombin-induced platelet aggregation in vitro. As shown in Figure 3D, treatment with both AHG100 and DAHGs inhibited blood platelet aggregation induced by thrombin in a concentration dependent manner. In particular, the platelet aggregation was significantly inhibited at 1 μg/mL of UFH, 25 μg/mL of LMWH, 10 μg/mL of AHG100, 50 μg/mL of DAHG50, and 50 μg/mL of DAHG10 (*p* < 0.05). These results indicate that AHG100 and DAHGs inhibit platelet aggregation through thrombin inactivation.

### 2.4. Effect of AHG100 and DAHGs on Clotting Time (CT) and Bleeding Time (BT)

BT and CT were applied to instinctively reflect the balance of anticoagulant effects and side effects. BT depends on the function of platelets and endotheliocytes, whereas CT is mainly affected by the presence of coagulation factors [21]. In Figure 4, CT in AHG100, DAHG50, and DAHG10 groups was visibly longer, with a Mw- and dose-dependent tendency compared with that in the NC group. DAHG50 and DAHG10 at a dose of 10 mg/kg increased the CT by 39.13% and 22.71%, respectively (*p* < 0.05). Regarding BT, intact AHG100 at the tested concentration and LMWH at 1 mg/mL exhibited similar increases in bleeding time compared with the NC group (*p* < 0.01). In contrast, administration of 1 and 10 mg/kg of DAHG10 did not obviously increase the hemorrhage risk (*p* > 0.05). Furthermore, CT/BT could reflect the relations between anticoagulant activities and bleeding risks. BT values of AHG100, DAHG50, and DAHG10 were only 88%, 59%, and 44% of that induced by LMWH, and these groups exhibited similar prolongation activities of CT at 10, 50, and 50 mg/kg, respectively. The CT/BT increased with Mw. This finding indicates that the therapeutic windows of DAHGs were wider than those of LMWH, suggesting that it is possible to enhance the therapeutic effects by elevating the dosage without increasing side effects to a certain extent.

### 2.5. Anti-Thrombus Action of AHG100 and DAHGs

The antithrombotic effects of AHG100 and DAHGs were assessed in an arterial thrombosis model. As shown in Figure 5, the short period of thrombus formation time and extensive thrombus formation in the NC group indicated that the model was successfully constructed. Thrombus formation time was prolonged along with the increasing concentration and Mw of glycosaminoglycans. The differences were significant compared with glycosaminoglycans in the measured range and the NC group (*p* < 0.05), even with 1 mg/kg of DAHG10. In addition, a dose-dependent and molecular-dependent antithrombotic action was observed after intravenous administration of AHG100 and DAHGs. This effect was characterized in obviously smaller thromboses in rats compared with that in the NC group. Specifically, DAHG10 inhibited thrombus formation by 61.7% and 65.4% at doses of 10 and 20 mg/kg, respectively, compared with that in the NC group (*p* < 0.05).

### 2.6. Correlation between Exposure and Anticoagulant Activity in Rats

The abovementioned experiments developed and validated the antithrombotic activity of AHG100 and DAHGs. Whether orally administered glycosaminoglycans possessed in vivo activity was studied. As shown in Figure 6, after intravenous injection with AHG100 and depolymerized products, Activated partial thromboplastin time (APTT) values reached up to 6.15, 3.22, and 2.03 times higher than that noted at the blank time, respectively (*p* < 0.01). Then, these values decreased with the extension of time. The effect is Mw-dependent. Thrombin time (TT) showed the same tendency as APTT, up to 2.16, 2.14, and 1.74 times longer than that of the blank time (*p* < 0.01). In addition, no obvious change in APTT and TT was noted after dosing with AHG100 and DAHG50 (intragastric gavage, i.g.). Oral DAHG10 yielded obvious increases in APTT and TT at 90 min (1.23 and 1.44 times that of blank time, respectively, *p* < 0.05). The reaction further reduced with the lapse of time.

Furthermore, the effects of oral DAHG10 on coagulation factors were determined. As shown in Figure 7B,D, the injection of DAHG10 suppressed the activity of FIIa and FXa at each sampling time (15–300 min). This effect gradually weakened over time. After oral administration, the residual activity of both FIIa and FXa was maximally decreased at 120 min (*p* < 0.05). Moreover, the effect of DAHG10 was more pronounced when a higher concentration of the oral anticoagulants was used. The dosage of 250 mg/kg of DAHG10 could significantly restrain the FIIa and FXa activity at all tested time points (*p* < 0.05). In addition, no factor XII (FXII) activation was observed when AHG100 and DAHGs were administered (Figure 7E,F), revealing their suitable safety profiles in vivo.

The plasma concentration of DAHG10 was represented by the concentration of disaccharide hydrolysate (CS-4S6S). As shown in Figure 8, DAHG10 could be detected at 30 min in rat plasma after oral administration at 250 mg/kg. Within 120 min, the concentration of DAHG10 in the plasma achieved a maximum value of 10.56 μg/mL. The pharmacokinetic parameters after oral dosing are listed in Table 2. The oral bioavailability of DAHG10 was calculated to be 1.16% according to the data presented in the previous report [22].

The correlation between concentration and efficacy was further established by describing the inhibition of FIIa and FXa activity that varies with time and plasma concentration. As shown in Figure 8, an overall positive correlation was noted between the blood concentration of DAHG10 and its inhibition ability. According to the PK-PD model fitting of the residual FXa and FIIa activity and concentration in Appendix A, the model parameters were calculated and are listed in Appendix A. Akaike’s information criterion (AIC) and Schwarz criterion (SC) values revealed that the model could well-fit the response curve of plasma concentration and inhibition activity. All R_obs-pre_ were 0.99, which means that the predicted results are in good agreement with the experimental values. The *E_max_* values for FXa and FIIa were 97.96% and 105.59%, respectively. The EC_50_ values of DAHG10 on FXa and FIIa inhibition were 57.59 and 50.75 μg/mL, respectively. For the intravenous injection group, *E_max_* was 101.59% for FIIa and 104.72% for FXa, and EC_50_ was 49.98 μg/mL for FIIa and 59.40 μg/mL for FXa.

## 3. Discussion

An obstructive thrombus propagates through the recruitment of platelets by thrombin and amplification of the coagulation cascade, eventually leading to deposition and maturation of a fibrin clot [23]. The coagulation cascade is the critical link in thrombosis. In this study, AHG100 was found to exert its potent anticoagulant effect on strong inhibition of intrinsic FXase complex, and AT-III-dependent inhibition of FIIa and FXa, which is consistent with the activities of other fucosylated chondroitin sulfates [10,11]. The structural consistency of depolymerization derivatives with intact AHG100 guaranteed the suppression of the intrinsic FXase complex, as well as to some extent Mw-dependently inactivated FIIa and FXa in the presence of AT-III. This result is in agreement with a previous study showing that the anti-FXase activity of DHG remained unaffected at a minimum size of approximately 8.5 kDa [11]. It was also reported that DHG with a molecular weight of less than 10 kDa exhibited enhanced selectivity in inhibiting the FXase complex with slight hemorrhage side effects [24,25]. It is likely that the intrinsic coagulation pathway could exert a crucial influence on pathological thrombosis rather than normal hemostasis [26]. The Mw-dependent tendency of BT reduction suggested that depolymerization was an efficient method to decrease the bleeding risk (Figure 4). Combined with in vitro and in vivo inhibitory effects on thrombin-induced platelet aggregation and thrombus formation, glycosaminoglycans with low Mw could pave a new way for the discovery and development of antithrombotic agents given their safety and efficiency.

It was hypothesized that the acidic gastric environment could reduce the effect of orally administered fucosylated chondroitin sulfate due to de-sulfation and de-fucosylation in previous reports [14,15]. However, we did not observe significant changes in free sulfate and Mw after AHG100 was incubated with artificial gastric fluid for 2 h (data not shown). This finding coincided with previous reports on the resistance of FCS from *Stichopus variegatus* and *Stichopus japonicus* to degradation in the gastrointestinal tract [27,28]. This phenomenon may occur because the high content of carboxyl and sulfate groups facilitated AHG100’s ability for pepsin inhibition. Hydroxyl groups in the uronic acid residues are thought to form hydrogen bonds with the active-site domain of pepsin [27]. Moreover, the absorption of DAHGs has been affirmed in intestinal cells [17]. Therefore, we assume that intact DAHGs can be absorbed. Then, the relationship between anticoagulant activity and absorption was tested for efficacy evaluation as an oral anticoagulant.

After oral administration, DAHG10 presented anticoagulant activity, which was characterized by a prolonged plasma coagulation time, and a dose-dependent decrease in FIIa and FXa activity. FII and FX are considered essential components in the intrinsic FXase to complete human blood clotting. FVIII (the component of the FXase complex) is thought to be activated by limited amounts of thrombin (FIIa) initially produced by FXa. Activated FVIII (FVIIIa) then amplifies FXa generation through intrinsic FXase complex, leading to increased FIIa production [29]. Therefore, intrinsic FXase complex activation could be quantified by measuring the inhibition rates of FIIa or FXa. Specifically, DAHG10 may dampen the intrinsic FXase complex to negate the activation of FXa and FXa in vivo. Regarding the antithrombotic mechanism, oral heparin with a similar Mw remains inactive, in accordance with previous research [30]. Moreover, the most notable side effects of fucosylated chondroitin sulfate were bradykinin release and severe hypotension induced by the activation of FXII [31]. Fortunately, activation of FXII was not observed when DAHG10 was administered orally. That is, DAHG10 could be effective orally, without risk for symptomatic hypotension.

The results were consistent with the previous research showing that DAHG with a lower Mw was more easily absorbed than AHG100 in the intestine transport model in vitro [17]. It could be assumed that DAHG10 could be absorbed through the intestinal barrier to the circulation, and hence influence coagulation function. A PK-PD model was further constructed to quantitatively describe the association between the antithrombin responses and the concentration of glycosaminoglycans. DAHG10 in plasma was quantified by disaccharides formed from the chondroitin sulfate-like backbone after intravascular injection [22]. In accordance with the plasma drug concentration–time curve, the absorption of DAHG10 was indicated by the C_max_ (10.56 μg/mL) and AUC_0-∞_ (5373.71 min·μg/mL) after dosing, providing evidence that orally administered DAHG10 was absorbed through the gastrointestinal tract to plasma. Based on the correlation study between exposure and anticoagulant activity in rats, the inhibition of FIIa and FXa was enhanced with the increasing DAHG10 concentration and peaked at 120 min in plasma. Anticoagulant activity is positively associated with the mean plasma concentration of DAHG10. Thus, although native AHG100 achieves more powerful anticoagulant and antithrombotic properties than DAHG10, the higher absorption efficiency of DAHG10 into the plasma can compensate for its weaker anticoagulant activity, resulting in a promising anticoagulant effect. However, due to the high polarity and poor lipophilicity of DAHG10, its oral bioavailability (1.16%) was low in vivo.

## 4. Materials and Methods

### 4.1. Materials

APTT and TT kits were purchased from Nanjing Jiancheng Bioengineering institute (Nanjing, China). FIIa, FXa, factor VIII (FVIII): C kit, and enoxaparin (LMWH, Mw_avg_ 5000 Da) were purchased from Hyphen Biomed (Neuville-sur-Oise, France). Chromogenic substrate S-2302 was obtained from Pentapharm (Basle, Switzerland). Human AT-III was purchased from Taibang Biological Product Co., Ltd (Taian, China). UFH (Mw_avg_ 14,000) was purchased from Sinopharm Chemical Reagent Co. (Shanghai, China). Dalteparin (LMWH, Mw_avg_ 5000) was obtained from Vetter Pharma-Fertigung GmbH & Co. (Ravensburg, Germany). Other used chemicals were of analytical reagent grade from Solabio (Beijing, China) and Sinopharm Chemical Reagent Co., Ltd (Shanghai, China).

Male Kunming mice (weight: 22–25 g), Wistar rats (weight: 220–250 g), and male Sprague Dawley rats (weight: 200–240 g) were provided by Qingdao Institute for Drug Control (Qingdao, China, SCXK20030010) or Jinan Pengyue laboratory animal breeding Co., Ltd. (Jinan, China, SCXK-20190007). Animals were acclimated for 7 days under a 12 h light/12 h dark cycle with free access to water and chow ad libitum at 25 ± 2 °C, 50–55% relative humidity. All animal care and experimental procedures were performed in accordance with China’s Guidelines on Welfare and Ethical Review for Laboratory Animals and approved by the Animal Ethics Committee of the School of Medicine and Pharmacy, Ocean University of China (Approval Code: OUC-SMP-2017-11-01).

### 4.2. The Preparation of AHG100 and DAHGs

AHG100 and DAHGs were prepared as described before [16,32]. Briefly, AHG100 was isolated and purified from the body wall of the fresh sea cucumber *Apostichopus japonicus*, as previously described [32]. DAHG50 and DAHG10 were prepared through controlled free-radical degradation with Cu^2+^ [16]. In detail, the intact AHG and CuAc_2_ (1 mmol/L) were dissolved in 50 mmol/L H_2_O_2_ to yield a final concentration of 1 mmol/L Cu^2+^. The reaction was continued for 2 and 6 h at 45 °C to obtain crude depolymerized products of DAHG50 and DAHG10, respectively. The derivatives were purified on a Q Sepharose Fast Flow column (300 × 30 mm) and a Sephadex 25 column (100 × 2.6 cm). The DAHG50 and DAHG10 were obtained by lyophilization after dialysis against distilled water over 72 h. Mw was determined by high-performance gel permeation chromatography (HPGPC) using a Waters e2695 HPLC (Waters, Milford, MA, USA) combined with a Waters 2410 refractive index detector (Waters, Milford, MA, USA). The samples (10 mg/mL) were dissolved in 0.2 mol/L Na_2_SO_4_, filtered through 0.22 µm membranes before injection, and eluted on a Ultrahydrogel™ linear Column (7.8 × 300 mm, 10 µm, Waters, Milford, MA, USA) with 0.2 mol/L of Na_2_SO_4_ at a flow rate of 0.5 mL/min. Then, Mw was calculated by reference to a calibration curve made by a series of dextran standards (Mw: 133.8, 84.4, 41.1, 21.4, 10.0, 7.1 kDa). ^1^H nuclear magnetic resonance (NMR) spectra were performed in D_2_O (99.9%, Sigma-Aldrich, Darmstadt, German) containing tetramethylsilane (TMS) as the internal standard at 25 °C using a DD2 500 MHz spectrometer (Agilent, Palo Alto, CA, USA). Fourier transform infrared (FT-IR) spectra were recorded on a Nicolet Nexus 470 spectrometer (Thermo Fisher Scientific, Waltham, MA, USA). Monosaccharide composition analysis was performed by a precolumn hydrolysis plus 1-phenyl-3-methyl-5-pyrazolone (PMP) labeling high-performance liquid chromatography (HPLC) method.

### 4.3. Effects of AHG100 and DAHGs on Coagulation (Co)Factors In Vitro

Coagulation (co)factor inhibition assays for the intrinsic FXase complex, FIIa, and FXa were performed according to the manufacturer’s recommended procedures with modifications. Briefly, the inhibition of intrinsic FXase complex was determined using the FVIII: C kit. AHG100 and DAHGs of various concentrations (0.001–1000 µg/mL) were incubated with 2 IU/mL of FVIII and 60 nmol/L of factor IXa (FIXa) at 37 °C for 2 min, followed by the addition of factor X (FX, 50 nmol/L ). After 1 min, 30 µL of 8.40 mmol/L FXa chromogenic substrate was added. Then, 20% acetic acid was employed to interrupt the reaction. UFH and LMWH served as positive controls. The inhibition of the intrinsic FXase complex was assessed by measuring absorbance at 405 nm using a Bio-Tek Microplate Reader (ELX 808, BioTek, Winooski, VT, USA). Moreover, the anti-FIIa and anti-FXa activities in the presence of AT-III (0.25 IU/mL for FIIa and 1 IU/mL for FXa activity, respectively) were measured according to the instructions supplied with the commercial kits. The chromogenic substrate for FIIa was S-2238 (1.25 mmol/L), and that for FXa was S-2765 (1.2 mmol/L). The optical density (OD) at 405 nm of the mixed solution was recorded and IC_50_ was calculated.

### 4.4. Effect of AHG100 and DAHGs on Thrombin-Induced Platelet Aggregation

Platelet-rich plasma (PRP) and platelet-poor plasma (PPP) were isolated by centrifugation of whole blood collected in tubes containing sodium citrate at 200× *g* for 5 min and 2000× *g* for 10 min. After the platelet count was adjusted with PPP, PRP was mixed with different concentrations of AHG100, DAHG50, and DAHG10. Then, the platelet aggregation rate with 5 min induction by FIIa (human thrombin, 1.25 NIH/mL) was assessed using an aggregometer (Precil, Beijing, China) according to Born’s method [33,34].

### 4.5. Effect of AHG100 and DAHGs on CT and BT

Male mice were divided into 11 groups: NC (normal saline) group, positive control (PC, dalteparin, 100 U/kg) group, and AHG100 and DAHGs (1, 10, 20 mg/kg) groups, and each group consisted of 10 animals. The assignment to groups was haphazard. The samples were intraperitoneally injected (i.p.) into mice at a single dose. After 30 min, the BT was measured using the tail breaking test, whereas the CT was recorded by the capillary test [35].

### 4.6. Anti-Thrombus Action of AHG100 and DAHGs in Rats

After 24 h of fasting, a total of 88 male rats were injected with a single dose of AHG100, DAHG50, and DAHG10, which were respectively dissolved in sterile normal saline at concentrations of 1, 10, and 20 mg/kg, through the femoral vein following anesthesia with 10% chloral hydrate. Rats from the NC group were treated with the vehicle under similar conditions, whereas PC rats were intraperitoneally injected with UFH (1 mg/kg).

Thirty minutes later, 1 mL blood samples were collected through aortaventralis and further infused into silicone thrombus tubes. The inhibition of thrombus formation was conducted on an MX-300 simulated blood clot instrument (Shanda industrial Co., Shanghai, China). The length and weight of the thrombus were recorded.

Moreover, the rats subject to the same administration approach were employed for the proximal arteria carotis communis thrombus assay induced by electrical stimulation (7.6 mA for 7 min). The antithrombotic activity was illustrated based on thrombus formation time using a BT87-3 experimental intracorporeal thrombosis surveyor with a temperature probe (designed by Cardiovascular Laboratory, BaoTou Medical College, Inner Mongolia University of Science and Technology, Inner Mongolia, China).

### 4.7. Correlation between Exposure and Anticoagulant Activity in Rats

Rats (male, *n* = 20) were orally administered (i.g.) AHG100, DAHG50, and DAHG10 at 50 and 250 mg/kg, and UFH at 50 mg/mL. Blood samples (0.3 mL) were collected from the external jugular vein cannula with syringes containing sodium citrate at 30, 60, 90, 120, 150, 180, 240, and 300 min. Blood volume was replenished with the same volume of normal saline following each sampling. The plasma was separated from the blood samples through centrifugation for 10 min at 4000 rpm. Intravenous administration was also performed in the i.v. group through the caudal vein.

APTT and TT were tested in accordance with the kit instructions. Moreover, the residual activity of FIIa, FXIIa, and FXa was evaluated using the chromogenic substrate method. In brief, 60 μL of a plasma and Tris buffer (containing Tris 0.05 mol/L, NaCl 0.175 mol/L, EDTA 7.5 mmol/L, PEG 0.1%, pH 8.4) mixture was incubated at 37 °C for 2 min, followed by the addition of chromogenic substrate S-2238 (FIIa, 1.25 mmol/L), S-2765 (FXa, 1.2 mmol/L), or S-2302 (FXIIa, 4 mmol/L). The residual coagulation factor activity was calculated based on the absorbance (405 nm) ratio of samples at each time point after administration to that of the blank control.

In addition, the quantification of DAHG10 in rat plasma was performed by following the approach proposed in our previous research with satisfactory sensitivity, accuracy, and precision [22]. Briefly, the plasma samples were prepared by digestion, mild acid hydrolysis, enzymolysis, and derivatization. One of the deuterogenic disaccharides of the main chain was analyzed using an UltiMate 3000 UHPLC system coupled with a TSQ QUANTIVA system (Thermo Fisher Scientific, Waltham, MA, USA) on a Cortecs UPLC T3 column (150 × 2.1 mm i.d., 1.6 μm). The pharmacokinetics (PK) analysis was performed by non-compartmental analysis using WinNonlin^®^ software (version 6.0, Pharsight^®^, a Certara™ Company, Princeton, NJ, USA). The PK parameters were characterized by the plasma drug concentration–time curve, including elimination half-life (T_1/2_), area under the plasma concentration time curve (AUC), maximum concentration in plasma (C_max_), time of maximum plasma concentration (T_max_), and mean residence time (MRT). The inhibitory effects of DAHG10 on FIIa and FXa, calculated with the following equation: inhibition (%) = 100 × (1− OD _sample_/OD _blank_), were used as the pharmacodynamics (PD) response in this study.

The PK-PD model was conducted to explain the correlation between exposure and anticoagulant activity of DAHG10 using PKsolver, an add-in program in Microsoft Excel [36]. Different pharmacodynamic model structures and model parameters were calculated to clarify the relationship between plasma concentration and inhibition of FIIa and FXa. The most appropriate *E_max_* model and weight were selected on account of the minimum AIC and coefficient of determination (*R*^2^). Among them, the inhibitory *E_max_* model for PK-PD analysis was selected based on the best fitness via the following equation:
*E* = *E**_max_* × [1 − *C*/(*C* + *E**C*_50_)](1)
where *E* (%) is the absolute inhibitory activity against FIIa or FXa, *E**_max_* (%) is the maximum possible effect that describes efficacy, *C* is the concentration in the hypothetical effective compartment, and EC_50_ (μg/mL) is the concentration at which 50% of the maximum inhibition effect was obtained. Here, the mean values (*n* = 4) of plasma concentration and residual FIIa (or FXa) activity (%) were used to profile the antithrombin effect versus time.

### 4.8. Statistical Analysis

Experimental values are expressed as the mean ± standard deviation (SD). Significant differences between groups were analyzed by one-way analysis of variance (ANOVA) with IBM SPSS statistics version 19.0 (IBM SPSS Inc., Chicago, IL, USA). *p*-values less than 0.05 were considered statistically significant.

## 5. Conclusions

The results showed that DAHGs, with intact primary structures compared with AHG100, retained anticoagulant activity, which includes the inhibition of the FXase complex activity, and antithrombin-dependent FIIa and FXa. They markedly prevented thrombosis formation and thrombin-induced platelet aggregation in a dose- and molecular weight-dependent manner in vitro and in vivo. Furthermore, DAHG10 exhibited obviously lower bleeding risks than native AHG100. Following oral administration, DAHG10 could be absorbed into blood, further dose-dependently prolonging APTT and TT and inhibiting FXa and FIIa partially through FXase. Anticoagulant activity was positively associated with plasma concentration following oral administration of DAHG10. Therefore, our study provides a direct link to clarify the relationship between exposure and the effect of fucosylated chondroitin sulfate for developing effective and safe oral antithrombotic agents.

## Figures and Tables

**Figure 1 marinedrugs-20-00631-f001:**
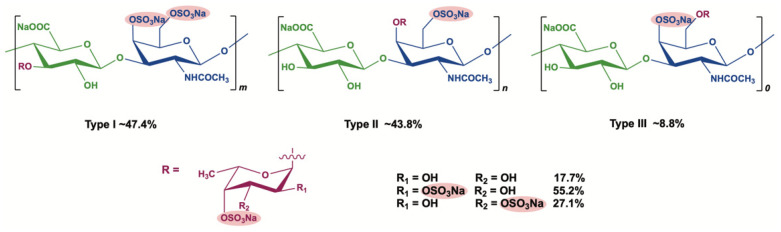
The structure of AHG100.

**Figure 2 marinedrugs-20-00631-f002:**
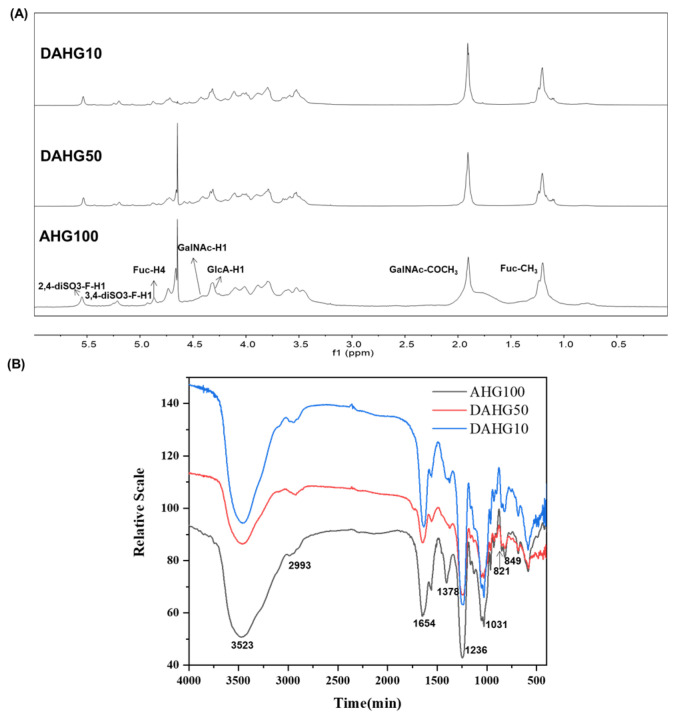
^1^H NMR and IR spectrum of AHG100 and DAHGs. (**A**) ^1^H NMR spectrum, (**B**) IR spectrum.

**Figure 3 marinedrugs-20-00631-f003:**
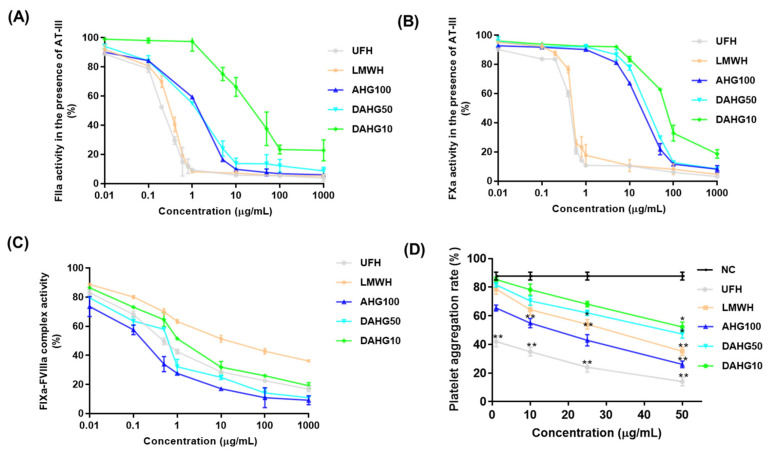
The structural information and anticoagulant activity of AHG100 and DAHGs (mean ± SD, *n* = 3). (**A**) Anti-FIIa activity, (**B**) anti-FXa activity, (**C**) anti-intrinsic FXase complex activity, and (**D**) platelet aggregation. Compared with normal control (NC) group, * *p* < 0.05, ** *p* < 0.01.

**Figure 4 marinedrugs-20-00631-f004:**
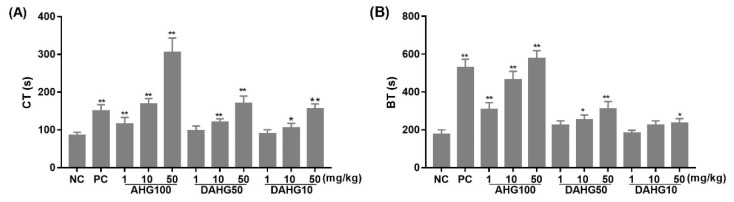
Effects of AHG100 and DAHGs on CT and BT (mean ± SD, *n* = 10). (**A**) CT, (**B**) BT. Compared with NC group * *p* < 0.05, ** *p* < 0.01.

**Figure 5 marinedrugs-20-00631-f005:**
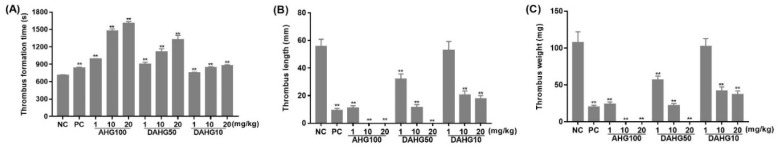
Effects of AHG100 and DAHGs on antithrombosis (mean ± SD, *n* = 8). (**A**) Thrombus formation time, (**B**) thrombus length, and (**C**) thrombus weight. Compared with NC group ** *p* < 0.01.

**Figure 6 marinedrugs-20-00631-f006:**
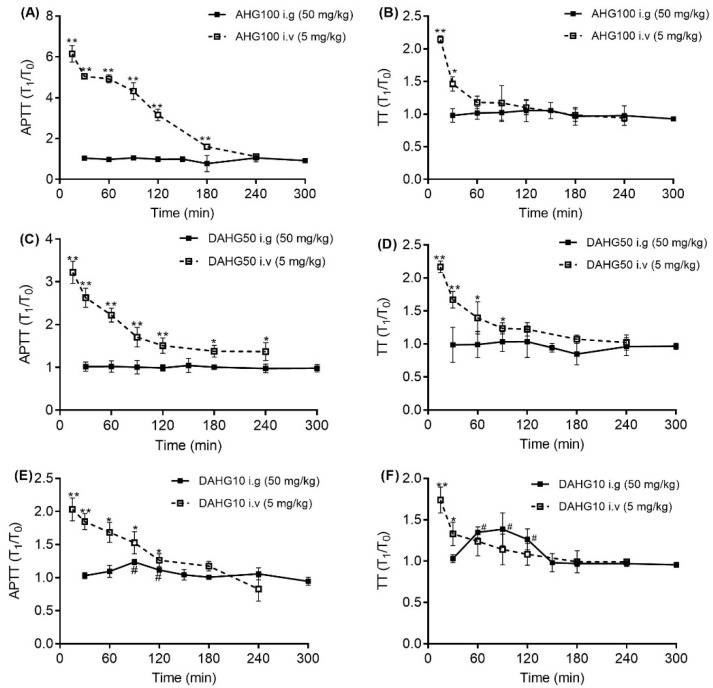
Effect of AHG100 and DAHGs on APTT and TT in rats (mean ± SD, *n* = 3). (**A**) AHG100, APTT; (**B**) AHG100, TT; (**C**) DAHG50, APTT; (**D**) DAHG50, TT; (**E**) DAHG10, APTT; (**F**) DAHG10, TT. Compared with blank time of intravenous administration, * *p* < 0.05, ** *p* < 0.01. Compared with blank time of oral administration, ^#^
*p* < 0.05.

**Figure 7 marinedrugs-20-00631-f007:**
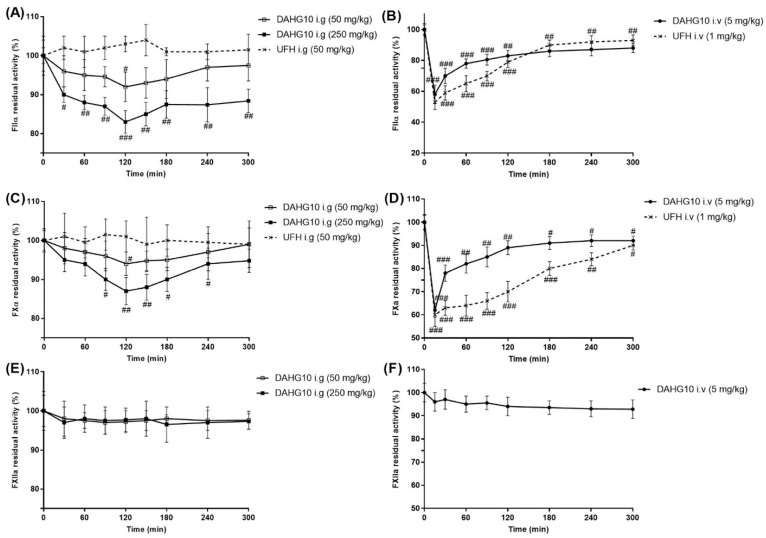
Effect of DAHG10 on FXa, FIIa, and FXIIa in rats (mean ± SD, *n* = 4). (**A**) Residual activity of FIIa after oral dosing with DAHG10. (**B**) Residual activity of FIIa after injection of DAHG10. (**C**) Residual activity of FXa after oral dosing with DAHG10. (**D**) Residual activity of FXa after injection of DAHG10. (**E**) Residual activity of factor XIIa (FXIIa) after oral dosing with DAHG10. (**F**) Residual activity of FXIIa after injection of DAHG10. Compared with blank time after administration, ^#^
*p* < 0.05, ^##^
*p* < 0.01, ^###^
*p* < 0.01.

**Figure 8 marinedrugs-20-00631-f008:**
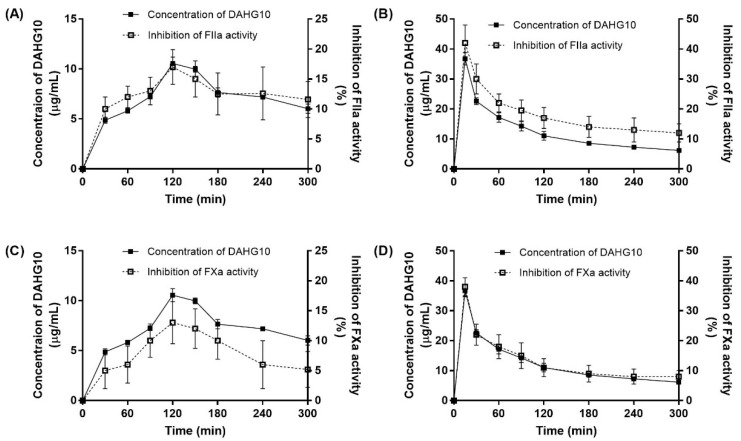
The concentration–time–effect relationship in rats (mean ± SD, *n* = 4). (**A**) Inhibition of FIIa activity after oral administration of DAHG10 at 250 mg/kg. (**B**) Inhibition of FIIa activity after intravenous injection of DAHG10 at 5 mg/kg [22]. (**C**) Inhibition of FXa activity after oral administration of DAHG10 at 250 mg/kg. (**D**) Inhibition of FXa activity after intravenous injection of DAHG10 at 5 mg/kg [22].

**Table 1 marinedrugs-20-00631-t001:** Monosaccharide composition analysis of AHG100 and DAHGs.

Compound	Mw (Da)	Chemical Composition (Molar Ratio)		
		GlcA	GalN	Fuc
AHG100	94,156	1.04	1.00	0.90
DAHG50	51,318	1.02	1.00	0.91
DAHG10	9972	1.02	1.00	0.94

**Table 2 marinedrugs-20-00631-t002:** Pharmacokinetic parameters of DAHG10 after oral administration at 250 mg/kg in rats.

Parameter	DAHG10
T_1/2_ (min)	367.92 ± 169.29
T_max_ (min)	120
C_m__ax_ (mg/mL)	10.56 ± 0.66
AUC_0–t_ (min·μg/mL)	2111.49 ± 39.75
AUC_0–∞_ (min·μg/mL)	5373.71 ± 1578.63
Vd (mL/kg)	23,655.92 ± 4718.83
MRT_0–t_ (min)	159.70 ± 1.07
MRT_0–∞_ (min)	571.78 ± 227.69
Cl (mL/min/kg)	50.08 ± 16.40

## Data Availability

All data, models, and code generated or used during the study appear in the submitted article.

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
