# Peer review of "In Vivo Anticoagulant and Antithrombic Activity of Depolymerized Glycosaminoglycan from Apostichopus japonicus and Dynamic Effect–Exposure Relationship in Rat Plasma"

_marinedrugs, 2022, doi:10.3390/md20100631_

Round 1

Reviewer 1 Report

It is possible an author is missing at the end of the author listing of the paper itself.  Otherwise there is a word that needs deletion.

In line 293 the work should be "provided" not "provide"

The authors need to detail the N of subjects used in each experiment as well as gender.

The authors need to describe how the animals were randomly divided?  Coin flip" random numbers generator? or was the assignment to groups haphazard, which is fine but just needs to be properly said.

The statistics used was likely not appropriate, though this is difficult to tell.  More than likely nonparametric statistics suitable for non-normal distributions should have been used.

No variation is shown on Figure S3 which seems unlikely.  Was N = 1?  The authors are encouraged to include more detais of the data used to create various figures.   Reviewer is assuming that more than one NMR sample was run, but there is no way to know.  Similar challenge throughout.

Author Response

Thank you so much for the comments. Our point-by-point responses are presented below.

  1. It is possible an author is missing at the end of the author listing of the paper itself.  Otherwise there is a word that needs deletion.

Reply: Thank you for pointing this out. We have deleted “and” at the end of the author sequence in line 5 in the revised manuscript.

  1. In line 293 the work should be "provided" not "provide".

Reply:We apologize for the spelling mistake. We have revised the "provide" to "provided" in line 311 in the revised manuscript.

  1. The authors need to detail the N of subjects used in each experiment as well as gender.

Reply:As you suggested, we have added the gender and amount of animals in the experiment section. You can see the change in line 367-369, 374 and 391 in the revised manuscript.

  1. The authors need to describe how the animals were randomly divided?  Coin flip" random numbers generator? or was the assignment to groups haphazard, which is fine but just needs to be properly said.

Reply:We gratefully appreciate your suggestion. The assignment to groups was haphazard. The description was added in line 369 in the revised manuscript.

  1. The statistics used was likely not appropriate, though this is difficult to tell. More than likely nonparametric statistics suitable for non-normal distributions should have been used.

Reply:We did the normal (Gaussian) distribution test. The data in our research accords with normal distribution. Therefore, one-way analysis of variance (ANOVA) was applied in statistics analysis.

  1. No variation is shown on Figure S3 which seems unlikely. Was N = 1? The authors are encouraged to include more detais of the data used to create various figures. Reviewer is assuming that more than one NMR sample was run, but there is no way to know. Similar challenge throughout.

Reply:Thanks for your comment. The mean values (n=4) of plasma concentration and residual FⅡa (or FⅩa) activity (%) was used to profile the antithrombin effect vs. time. We have provided details in line 432-433 in the revised manuscript. Speaking of NMR, the data of DAHGs was similar to those of native AHGs, which indicated that primary structure of DAHGs did not change during the depolymerization. The controlled Cu2+ catalytic peroxide depolymerization process was a common and convenient degradation method, so it was widely used in the structure determination of native FCS from various sea cucumber species (Carbohyd. Polym., 270 (2021), 118368; Eur. J. Med. Chem., 92 (2015), 257-269). In particular, previous results showed that the primary structure of peroxide depolymerized products was similar to those of native FCS (Food Chem., 122 (3) (2010), 716-723; Tetrahedron Letters, 33 (1992), 4959-4962). The previous research work of our lab also demonstrated no significant variations of the backbone and branches structures during the depolymerization (Int. J. Biol. Macromol., 72 (2015), 699-705). Therefore, there were no significant variations detected in the NMR and IR spectra between AHG100 and DAHGs. You can see the statement in line 96-106 in the revised manuscript.

Reviewer 2 Report

This is an interesting manuscript which addresses anticoagulant and antithrombotic activity of depolymerized glycosaminoglycans of marine source -  AHG. The authors may consider the following editorial comments:

1. Please include a separate materials and methods section detailing the methods used for both in vitro and in vivo studies. 

2. How was the AHG depolymerized? 

3. How was the IC50 for ATIII supplemented systems determined for both the anti-Xa anti-IIa activities.

4.  Is there any comparative data with heparin or LMWH?

5. Please include the materials and methods before the results.

6. Do you have any structure data on the AHG?

Author Response

Thank you so much for the comments. Our point-by-point responses are presented below.

  1. Please include a separate materials and methods section detailing the methods used for both in vitro and in vivo studies.

Reply:Thanks for your kind suggestion. The materials and methods section was placed in line 298 to 440 in the revised manuscript.

  1. How was the AHG depolymerized?

Reply:The Cu2+ catalyzed free-radical depolymerization process was applied to prepare the low molecular weight glycosaminoglycan without destroying the primary structure of polysaccharide. We have added the corresponding description in line 324 to 343 in the revised manuscript.

  1. How was the IC50 for ATIII supplemented systems determined for both the anti-Xa anti-IIa activities.

Reply:The main physiologic function of AT-Ⅲ is to inactivate activated coagulation factor X (FXa) and thrombin (FIIa). AT-Ⅲ is a progressive inhibitor and the rate of its reaction with the active coagulation factors is slow, but in the presence of heparin or heparan sulfate, the rate of inhibition is accelerated 500- to 1,000-fold. It has been well demonstrated that the anticoagulant effect of heparin-like glycosaminoglycan could be mediated through its interaction with AT-Ⅲ (Thromb. Res., 146 (2016),59-68; Mar. Drugs 14(9) (2016), 170). In this work, we found that the inhibitory effects of AHG100 and DAHGs on Fxa and FIIa were AT-Ⅲ-depedented.

  1. Is there any comparative data with heparin or LMWH?

Reply:In our research, we compared the effect of AHG and DAGHs with that of heparin or LMWH on clotting time (CT), bleeding time (BT), and antithrombus action. Moreover, the effect of heparin by oral administration on blood coagulation was also investigated. However, oral heparin remains in-active. Therefore, the effect-exposure relationship research was only studied for describing the anticoagulant activity and plasma concentration of orally available AHG and DAHGs.

  1. Please include the materials and methods before the results.

Reply:Thank you for the comment. The order of sections is required by Marine Drugs.

  1. Do you have any structure data on the AHG?

Reply:Yes, we have. The structure data of AHG and DAHGs was shown in Table 1, Figure 1 and 2 in the revised manuscript.